# A time-domain phase diagram of metastable states in a charge ordered quantum material

Jan Ravnik[1,2], Michele Diego[1], Yaroslav Gerasimenko [1,3], Yevhenii Vaskivskyi [1], Igor Vaskivskyi [1,3], Tomaz Mertelj[1,3], Jaka Vodeb[1] & Dragan Mihailovic [1,3,4✉]

Metastable self-organized electronic states in quantum materials are of fundamental importance, displaying emergent dynamical properties that may be used in new generations of sensors and memory devices. Such states are typically formed through phase transitions under non-equilibrium conditions and the final state is reached through processes that span a large range of timescales. Conventionally, phase diagrams of materials are thought of as static, without temporal evolution. However, many functional properties of materials arise as a result of complex temporal changes in the material occurring on different timescales. Hitherto, such properties were not considered within the context of a temporally-evolving phase diagram, even though, under non-equilibrium conditions, different phases typically evolve on different timescales. Here, by using time-resolved optical techniques and femtosecond-pulse-excited scanning tunneling microscopy (STM), we track the evolution of the metastable states in a material that has been of wide recent interest, the quasi-two-dimensional dichalcogenide $1T$-TaS$_2$. We map out its temporal phase diagram using the photon density and temperature as control parameters on timescales ranging from $10^{-12}$ to $10^3$ s. The introduction of a time-domain axis in the phase diagram enables us to follow the evolution of metastable emergent states created by different phase transition mechanisms on different timescales, thus enabling comparison with theoretical predictions of the phase diagram, and opening the way to understanding of the complex ordering processes in metastable materials.

[1] Complex Matter Department, Jozef Stefan Institute, Ljubljana, Slovenia. [2] Laboratory for Micro and Nanotechnology, Paul Scherrer Institut, Villigen PSI, Switzerland. [3] Center of Excellence on Nanoscience and Nanotechnology—Nanocenter (CENN Nanocenter), Ljubljana, Slovenia. [4] Department of Physics, Faculty of Mathematics and Physics, University of Ljubljana, Ljubljana, Slovenia. ✉email: dragan.mihailovic@ijs.si

Short optical or electrical pulses can create non-equilibrium conditions that lead to electronic self-organization in quantum materials which can be usefully applied in various devices[1–9]. Recent rapid progress in the time-domain investigations of non-equilibrium phase transitions has led to the observation of a variety of emergent transient and metastable states in complex quantum materials, including organic electronic crystals[10], oxides[11,12], dichalcogenides[13], and fullerene superconductors[14]. A self-organized non-equilibrium matter that emerges after pulsed laser excitation evolves in a sequence of processes that eventually cause it to reach the ground state. However, sometimes new emergent metastable states may be created during the relaxation process which is not present in the equilibrium phase diagram[13].

The formation of such metastable states can occur by different mechanisms: in addition to first or second-order transitions characterized along the Ehrenfest[15] scheme, topological[16], and jamming[16,17] transitions may occur under non-equilibrium conditions—leading to fundamentally different dynamical ordering phenomena and diverse spatial orders. For example, domain structures can emerge through the Kibble–Zurek[18,19] mechanism in second-order transitions or transient mesoscopic phase separation in first-order transitions. To obtain a broader understanding of the dynamics of mesoscopic emergent phenomena that accompany such phase transitions the boundaries between metastable phases need to be determined on different timescales.

Presenting phase diagrams of temporally-evolving systems is a significant challenge. Most conventional phase diagrams focus on the study of equilibrium states, which means the system does not change much on a laboratory timescale of days, weeks, or a few years, i.e., over a timescale of $10^4$–$10^8$ s. This anthropocentric range of timescales is actually small compared to the >12 orders of magnitude that we present here. Our premise is that we need to make sure that the measurement time-window is short enough to perceive the system as approximately static (in quasi-equilibrium) on that respective time scale. The different techniques place our measurements into different temporal windows.

Scanning probe microscopy, which gives detailed microscopic and mesoscopic information works well on timescales $10^{-1}$–$10^3$ s. On the other hand, ultrashort timescale phenomena require stroboscopic repetitive scanning, which requires that the system dynamics is periodic, i.e., the transition outcome should be the same each time, so that it can be probed many times at the same moment in its cycle, and the signal can be averaged over many cycles. This has significant limitations, such as the requirement that all phenomena need to fully relax between measurements. The outcomes of singular ultrafast events, such as ultrafast phase transitions caused by a single laser shot that lead to metastable states which are of particular interest here, are an even greater challenge. Single-shot techniques are limited by balancing signal-to-noise and damage by the probe unless the metastable state lifetime is sufficiently long, to allow examination by STM or AFM for example. We should also keep in mind that states created under non-equilibrium conditions may result in different outcomes, depending on the experimental conditions, such as the quality of thermal and mechanical contacts, film thicknesses, macroscopic experimental geometry, the thermal pathway, etc., which need to be carefully controlled for a particular realization of the phase diagram.

Yet, as we show here, using a combination of multi-pulse femtosecond time-resolved coherent phonon spectroscopy in combination with STM, a particular temporal phase diagram may be pieced together when the lifetime of the states can be tuned by temperature The different phases that appear in the experiment on different timescales can then be compared with the results of a theoretical charged-lattice-gas model calculation of equilibrium and metastable photoinduced phases.

The studied material is a prototypical quasi-2D transition metal dichalcogenide TaS$_2$, which was shown recently to display multiple metastable electronic and structural ordering phenomena. It is a layered material, consisting of S–Ta–S trilayers, bonded together with weak Van-der-Waals chemical bonds, giving the material a strong 2D character. Its phase diagram includes different structural polytypes (1T-TaS$_2$, 2H-TaS$_2$ etc.)[20], different configurational charge-ordered states[13,16,17,21,22], superconductivity[23] and a quantum spin liquid candidate state[24].

The 1T polytype has an orthorhombic unit cell and is metallic above 550 K, without any charge modulation. In the range 350–550 K it displays an incommensurate (IC) charge density wave (CDW), which undergoes a transition to a nearly-commensurate (NC) phase below ~350 K. The NC state consists of ordered domains that are separated with rather broad smoothed out domain walls. Below 180 K, the material becomes insulating and fully commensurate (C) with a $\sqrt{13} \times \sqrt{13}$ superlattice structure, discussed either in terms of a commensurate CDW, a polaronic Wigner crystal[25–27], or a Mott state[23]. The material also shows low-$T$ superconductive behavior under high pressure[23]. Upon heating from the C state, the material goes through a triclinic stripe domain state in the range 220–280 K, whereupon it reverts to the NC state. Among the many different equilibrium orders, the material also exhibits long-lived metastable states. Of particular interest are (i) the non-equilibrium topological transition to a hidden (H) metallic state with chiral domain structures[16], and (ii) the jamming transition to an amorphous (A) state with hyperuniform electronic order[17]. Both states show electronic ordering, which cannot be found in equilibrium but can be reached by photoexcitation or electrical excitation.

The 2H polytype is the thermodynamically stable polytype with a trigonal prismatic unit cell and shows a much different phase diagram than the 1 T polytype. It is metallic above $T = 75$ K. Below 75 K, it forms a commensurate CDW state with $3 \times 3$ superlattice[28]. It becomes superconducting below $T = 0.8$ K[29].

The non-equilibrium phases in 1T-TaS$_2$ were previously investigated by optical pump-probe techniques[30], transport measurements[1,3,4,31], time-resolved X-ray diffraction[32–34], and time-resolved electron diffraction (TrED)[35–38], sketching the timeline of the transient phenomena. After the creation of $e$–$h$ pairs by the incident laser photons (in <1 fs), melting of the Mott state takes place within ~50 fs, while the periodic lattice modulation melts on the timescale of the collective mode 1/2-period (~200 fs)[39]. The transition to the H state was observed by coherent phonon measurements to take place in ~450 fs[30]. TrED measurements[35] revealed a number of transient phases on the picosecond timescale dependent on photon density, with a semi-continuous rotation of the CDW ordering wavevector with respect to the crystal axes as a function of laser fluence. The ordering wavevector angles observed in TrED are consistent with the angles obtained from the Fourier transformed STM images[16] and with recent static X-ray measurements of the H state[40]. X-ray diffraction showed electronic domain fusion dynamics associated with the formation of the IC phase, described in terms of coherent domain formation on the picosecond timescale[38] followed by diffusive processes on a time-scale of ~100 ps[34].

Importantly for this study, the lifetime of the hidden (H) state is temperature-dependent, ranging from an extrapolated ≫$10^{10}$ s at 4 K to <$10^{-4}$ s at 150 K, which allows us to tune its relaxation time[2], facilitating stroboscopic measurements. The A phase was suggested to be stable to temperatures above 200 K[17] but the temperature-dependence of its lifetime is not known.

## Results and discussion

To map out the temporal phase diagram we link STM images of different phases with the femtosecond optical spectroscopy results. First, we identify which long-lived nonequilibrium phases we can reach with different photoexcitation densities. To this end, we conduct STM experiments at low temperatures (4 K), where all of the previously known photoinduced phases can be considered stable, and we can distinguish them by their distinct spatial electronic ordering. Next, we link the latter to specific coherent phonon spectral features measured with the low-fluence two-pulse transient reflectivity spectroscopy. With the phonon fingerprints of the various states ascertained, we present three pulse technique measurements designed to track the non-equilibrium phonon evolution on short timescales.

We use an ultrahigh vacuum (UHV) low-temperature STM (LT Nanoprobe, ScientaOmicron) with optical access (Fig. 1a). The $1T$-TaS$_2$ samples are excited with an external laser. Since the STM scans cover a very small ($30 \times 30$ nm$^2$) area of an elliptical Gaussian laser spot ($\sim100 \times 150$ μm$^2$ at FWHM) we can investigate the effect of different laser fluences simply by choosing an appropriate area of the STM scan with respect to the center of the Gaussian beam. The excitation is done with a laser pulse with the peak fluence of 12 mJ/cm$^2$, which gives us a possibility to perform STM measurements over a wide range of fluences $F = 0$–$12$ mJ/cm$^2$ (Fig. 1b). Typical experiments are done with a single laser pulse photoexcitation, but multiple pulse excitations were also performed to understand the effects of heating the sample. The outcome of 1 and 10$^6$ shot experiments are shown in Fig. 1c, d, where the colors of triangles represent the different observed states (Fig. 1e–j). In both cases, the system returns to the original (C) state after photoexcitation (Fig. 1e, blue in c, d) in the region where the fluence $F < 1.5$ mJ/cm$^2$. Above $\sim1.5$ mJ/cm$^2$ the characteristic H state domain mosaics[16] consistently appears (Fig. 1f, yellow in c, d), independent of the number of the excitation pulses. Increasing the fluence beyond $\sim3.5$ mJ/cm$^2$ patches of the amorphous A state with a characteristic hyperuniform electronic structure[17] start to appear within the H-state areas (Fig. 1g, pink in c, d). These appear very inconsistently, and independent of the number of excitation pulses. At even higher fluences, we observe irreversible structural changes (ISC) that cannot be annealed by heating the sample (black). These include single layer polytype transformations from 1T to 1H (Fig. 1h), which appear as triangular patches of various sizes. The 1H polytype can be clearly distinguished from the 1T polytype based on the lack of charge ordering above 75 K and by a $3 \times 3$ CDW at lower temperature[20]. Another form of ISC is the layer peel-off and the creation of 1D nanotube-like objects (Fig. 1i). Eventually, melting/ablation of the material is observed and an uneven surface is created with nanoscale modulations (Fig. 1j). In the extreme cases of fluences well above 10 mJ/cm$^2$ or very thin flakes with poor connection to the bulk, craters a few microns to hundreds of microns in diameter can appear (Fig. 1b). Large regions of ISC sometimes appear in single pulse experiments, but are more common with pulse-train excitation (Fig. 1d) (Fig. 1c shows an example of a single pulse excitation, where ISC did not appear, however, this is not always the case), implying that the accumulated heating of multiple pulses enhances the formation of ISC. The ISC areas are surrounded and sometimes intermixed either by H or A state on micrometer scale regions. We further note that for high fluences >3 mJ/cm$^2$ the transition outcome is not perfectly reproducible on different areas of the sample. We attribute this to the uneven thermal coupling between the sample and the substrate, possible imperfections of the layer stacking, or strains caused during cleaving.

Having established the boundaries of the phase diagram on the STM timescale of $\sim10^3$ s, we use femtosecond spectroscopy with

three different pulse sequences (Fig. 2a) to investigate the trajectory of the system on the timescale down to $10^{-13}$ s via coherent phonons: (i) a single driving (D) pulse causes the change of state, followed by a standard stroboscopic two-pulse pump (P)–probe (p) experiment, in which the timescale is defined by the total time of the measurement ($\sim10$ min). The weakly-perturbative P–p protocol allows us to extract the near-equilibrium phonon fingerprints of the metastable phases on the same timescales and temperatures as in the STM experiments, thus matching the signatures obtained by the two techniques. (ii) A repetitive three-pulse D–P–p sequence gives access to the picosecond timescale. Here, each scan of the P–p experiment is taken with a small (ps) fixed delay between the D and P pulses. This is applicable for mapping the H state at temperatures where its lifetime is shorter than the repetition time (1 ms using a 1 kHz laser system) and thus the sample fully relaxes before the next D–P–p sequence arrives. (iii) The D–P–p technique, with the changed order of the pulses. Here the P–p pulses come a few tens of ps before the D pulse, which effectively produces $\sim1$ ms D–P delay (defined by the repetition rate). This way, we are able to establish whether the sample has relaxed between the pulses, or in the case that it has not relaxed, we are able to see which state was present after 1 ms. To ascertain that no permanent change has occurred in the sample in the case when it does not completely relax in 1 ms, we turn off the D pulse and re-measure the reflectivity transient using only the weakly perturbative P–p sequence.

The C and H states can be distinguished by the frequency of the collective amplitude mode (AM)[13]. While the AM frequencies in both states are temperature dependent, the AM frequency of the H state is about 0.1 THz lower than the AM frequency of the C state at any given temperature[30], thus making the two states easy to distinguish. The T and NC states show a characteristic double peak in the phonon spectrum with lower amplitude and about 0.1 THz lower frequency than the H state at any temperature and can thus also be clearly differentiated[30]. In Fig. 2a and b we show the transient reflectivity oscillations and the respective Fourier spectra for C, H, and T states. A detailed analysis of the C, H, NC, and T state AM peak frequencies and line shapes with multi phonon fits and their temperature dependences are given in Ref. [28].

The data for different D pulse fluences $F = 0.2 \sim 10$ mJ/cm$^2$ were taken at 80, 100, 140, 160, and 200 K. In Fig. 2c, d we show the transient reflectivity oscillations and the respective Fourier spectrum at 160 K at different D fluences after 30 ps and after a millisecond after switching to the metastable state.

Following the difference between the AM peaks in the C and H state, the C/H boundary on the low-fluence side can be defined with a high degree of certainty. The photoinduced H state at fluences slightly above 1 mJ/cm$^2$ forms already on the picosecond timescale and relaxes at this temperature within the $\sim1$ ms time between the pulses. The same was observed at 100 K and 140 K. At 80 K, the H state does not completely relax between the successive pulses, which is in agreement with the STM data at 77 K.

On the high-fluence side, the boundary of the H state cannot be as clearly ascertained. With increasing fluence, we see lowering of the AM peak intensity, broadening of the peak, and a further shift to lower frequencies. In this regime, we cannot easily recognize a unique fingerprint of any of the known states. To ascertain the presence of the A state, we compare the spectra of pristine and exposed samples 1 ms after photoexcitation at high temperatures, where the contribution of the H phase is absent due to its short lifetime. We see that for fluences above $\sim3$ mJ/cm$^2$ the sample does not relax completely to the C state in 1 ms, which is expected for the A state, but may also appear when other unidentified long-lived disordered phases or phases separation is

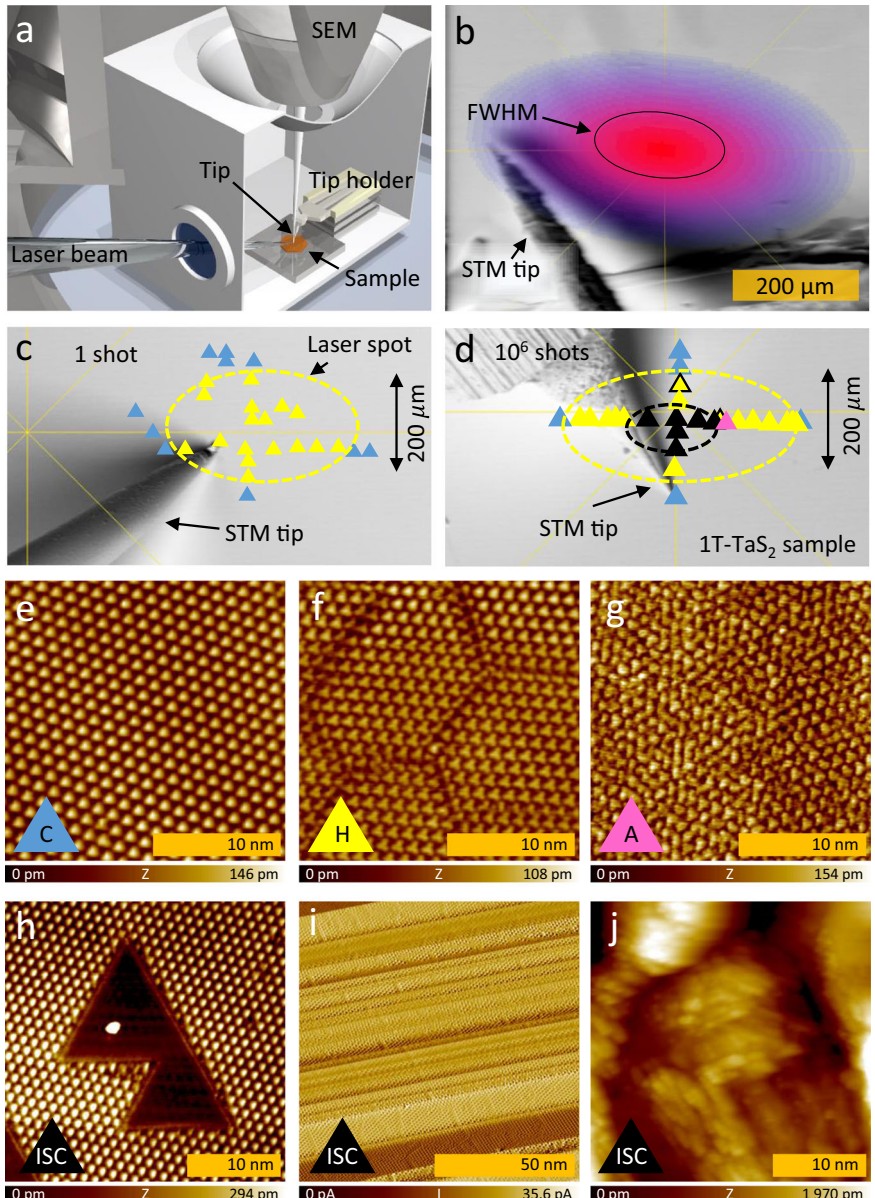

**Fig. 1 The STM experiment. a** A scheme of the optically excited scanning tunneling microscope (STM) setup with a vertically positioned scanning electron microscope (SEM). **b** SEM image of an area continuously exposed to 22 mJ/cm$^2$ peak fluence, showing damage at the center of the laser spot, which was used in the initial optical setup to define the beam position. The local fluence decreases as we move away from the center of the beam (from red to blue as marked on the image). The full width half maximum (FWHM) of the beam is marked with a black ellipse. **c, d** SEM images of the sample with the marked state profile (colors of the triangles represent different states as defined in (**e**)–(**j**)) after excitation with a single shot and multiple shots respectively. The colored ellipses serve only as a guide to the eye. **e–j** STM images of various states achieved by photoexcitation at 5 K. **e** commensurate (C) state, with a perfect triangular lattice of polarons (bright spots), (**f**) hidden (H) state with domains of ordered polarons, (**g**) amorphous (A) state with hyperuniform polaronic order, (**h**), (**I**), and (**j**) different examples of irreversible structural changes (ISC), namely (**h**) polytype transformation (at 77 K), where dark regions represent the 1H polytype monolayer which has no charge modulation at this temperature[20] (**i**) 1D surface ripples and (**j**) sample melting.

present. As observed by STM for $F < 7$ mJ/cm$^2$, such transient states with no characteristic oscillation fingerprint eventually evolve into either the H or the A state. Since (i) multiple pulse experiments in STM show little or no difference from single pulse experiments (with fluences below the damage threshold) and (ii) repetitive single-shot measurements in the same spot in STM show different electronic ordering pattern of the H and A states after each pulse, we conclude that the final photoinduced states are reconfigured by each pulse and independent of the initial electronic order.

For $F > 7$ mJ/cm$^2$, the AM peak intensity decreases even further, indicating the appearance of ISC. The irreversibility of the

excitation process was tested by turning off the D pulse and measuring a control P–p transient from the same spot. At fluences up to 7 mJ/cm$^2$, the signal mostly recovers. With increasing the fluence beyond 7 mJ/cm$^2$, the signal recovers only partially, while finally, at fluences above 10 mJ/cm$^2$, the sample suffers enough damage to completely suppress the signal, which is consistent with the STM scans.

The time-domain phase diagram with a compilation of the transition outcomes for different $F$ and $T$ is shown in Fig. 3, combining STM (triangles) and transient reflectivity data (squares). The time-axis signifies the time after the driving pulse (Fig. 3a) grouping the data into three timescales: ultrashort

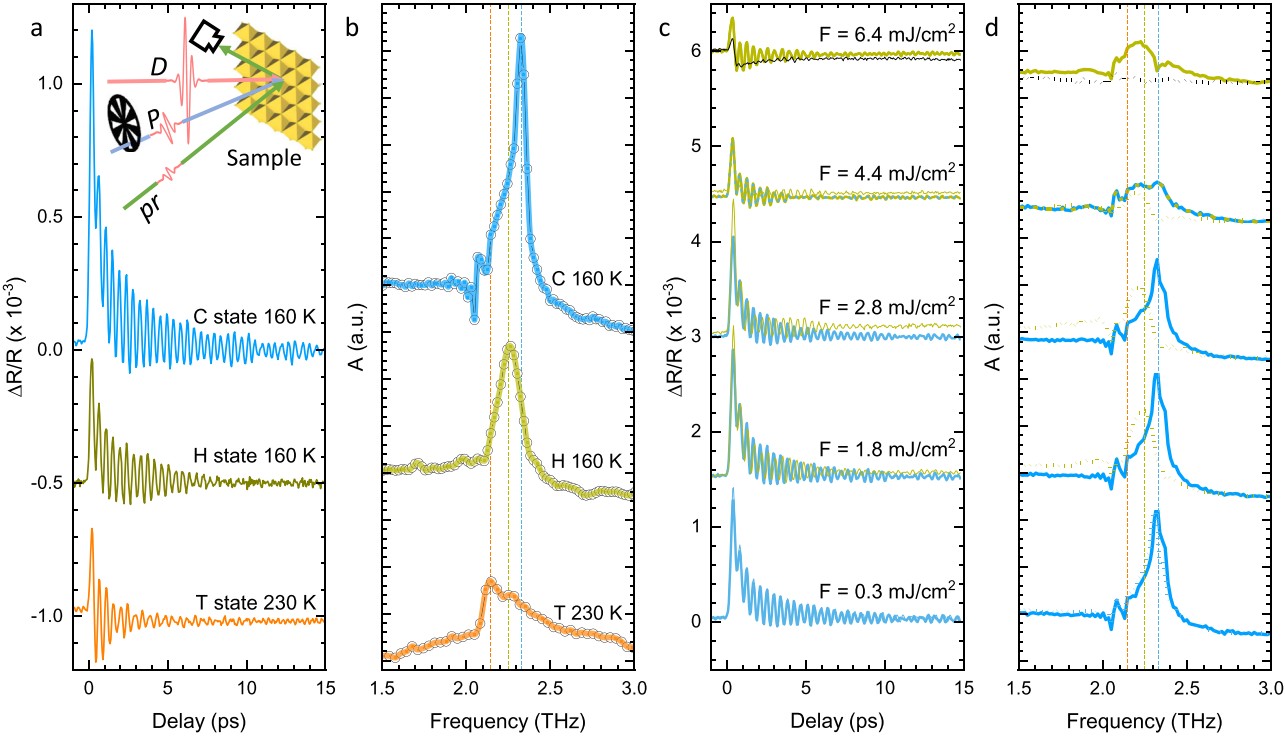

**Fig. 2 Transient reflectivity data. a, b** The transient reflectivity oscillations and their Fourier spectra for commensurate, hidden, and triclinic (C, H, and T) state, measured with three-pulse (C and H states) and two-pulse (T state) technique. The inset in (**a**) shows the schematic of the setup. The vertical dashed lines in (**b**) mark the peaks and are also shown in (**d**) for comparison. **c** Three-pulse drive-pump-probe (D–P–p) transient reflectivity measurements for different fluences at 30 ps (bold lines) and 1 ms (thin lines) showing different transition outcomes at 160 K, (**d**) their respective Fourier transforms. The bold and dotted spectra correspond to the measurements at 30-ps and 1-ms, respectively.

($<10^{-10}$ s), intermediate ($10^{-3}$ s), and long ($10^3$ s). The F axis is converted into the photon density taking a penetration depth of 30 nm[13] at 800 nm (Fig. 3b). The color shade (blue, yellow, pink, black) represent different states as shown in Fig. 1e–j. White represents unidentified transient states, or an inhomogeneous mixture of states, which cannot be separated spectroscopically.

We see that the H state has a nearly temperature-independent threshold fluence and is stable at low temperature in agreement with Ref. [3]. When the measurement timescale is longer than the relaxation time, the H state disappears, thus the C/H boundary moves to lower temperature with increasing time. The phase boundary of the C state observed by TRED[35] agrees remarkably well with the present measurements on short timescales. Also, the suppression of the CDW diffraction peaks is in agreement with the decrease of the AM peak intensity in the transient optical spectroscopy data reported here[30,35]. Finally, we note that the H state boundaries on long timescales are also consistent with the recent XRD measurements[40]. Comparing the appearance of the H state on short and long timescales we see that the H state appears on the ps timescale only at low fluences, but at higher fluences, it stabilizes later, on the STM timescale. Another observation, which illustrates the H state relaxation dynamics is that at 77 K the H state is only observable on the STM timescale at fluences above ~5 mJ/cm², while at lower fluences it is observed only on the ps timescale.

To obtain insight into the origin of different phases we compare the observed experimental phases on the STM timescale with the equilibrium configurational states obtained from theoretical treatment. The model considers ordering of immobile polarons subject to screened Coulomb interaction on a triangular atomic lattice and was previously successfully applied to describe both irregular domain patterns[25,26] and hyperuniform polaron

orders[17,25] in the H and A states, respectively. Its predictions can be compared with the experiment by assuming a correspondence between the photoexcited carrier density (which is proportional to incident photon density) and electron filling. The model defines the filling of the system as the number of polarons per unit cell[25]. In $1T$-TaS$_2$ with one electron per Ta atom, a $\sqrt{13} \times \sqrt{13}$ reconstruction of the CCDW state gaps 12 out of 13 electrons, resulting in 1/13 filling by the remaining electron[41,42]. Doping is defined as a change of the filling with respect to 1/13. Experimentally, the filling is obtained by counting the number of polarons per unit cell in an STM image (1 polaron equals 1 electron)[17,25].

Monte-Carlo simulations using this model give a theoretical phase diagram (Fig. 3c) that is consistent with the experimentally observed C (1/13 filling) and H states at ~4% nominal doping[25,26], observed at the experimental photodoping of 0.09 photons/unit cell. Remarkably, it also predicts the A state towards 1/11 filling (at ≳15% nominal doping, observed at a threshold of ~0.3 photons/unit cell)[17,25]. Previous simulations[25] for a larger range of fillings (from ~1/2 to ~1/21) predict the existence of various commensurate, domain, and amorphous states. Many, but not all, are present in various other materials[25]. It is interesting to note that even though we are exciting the samples with a continuous range of fluences, only certain values of polaron fillings are observed experimentally. To understand the physics behind this observation, let us consider the processes that take place during equilibration.

After the photoexcitation of the C state, an equal number of electrons (e) and holes (h) is created, whose density is proportional to the incident fluence. ARPES measurements show that the carriers thermalize within <300 fs[43]. Because of the strong e–h asymmetry of the band structure, the e and h thermalize at

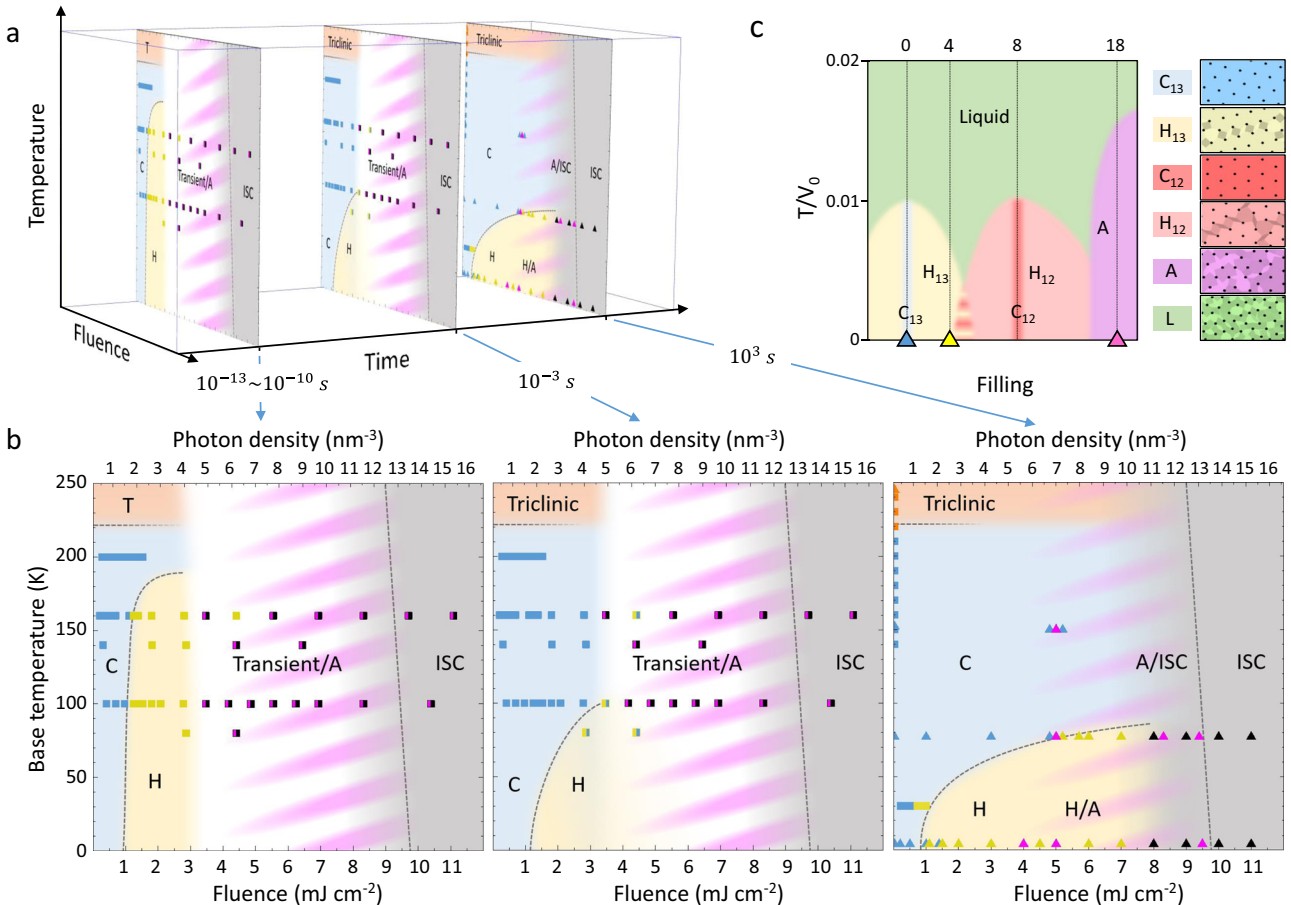

**Fig. 3 The time-domain phase diagram. a** Schematic timeline of the phase diagram evolution. **b** Detailed phase diagrams on the three timescales, as indicated. Triangles represent the scanning tunneling microscopy (STM) data and squares represent the transient reflectivity data (color code from Fig. 1e–j, and white represent the unresolved transient states). Note the significant shift of the long-time H state stability at high temperatures to higher fluences. **c** The equilibrium phase diagram obtained from simulations[25]. The C phase (1/13 filling-blue), the H phase (around 1/13 filling-yellow), and the A phase (~1/11 filling-violet) are all observed experimentally. The predicted phases at and around 1/12 filling (red) are not observed in any of the experiments. The green area represents the high-temperature liquid phase. The triangles and the dashed lines show at which filling values we observe the various states by STM.

different rates, leading to a transient imbalance of the carrier population at the Fermi level, which is usually referred to as photodoping[13]. As thermal equilibrium is approached, the carriers cause the lattice to respond on a timescale of ~1/2 of the collective vibrational mode period (<250 fs)[37], and polarons start to form, as mobile excitations without any long-range order. The polarons may either exist in a liquid state at high temperatures or form some kind of order (C, H, or A) at low temperatures (note that, for the sake of simplicity, this discussion deliberately ignores the existence of other ordered phases at higher temperatures).

Comparing this process with the theoretical model predictions at higher temperatures, the model shows a liquid state, where the polarons move around due to high temperature[25]. This liquid state is suggested to be closely related to the behavior of the material directly after photoexcitation. We can thus tentatively relate the unidentified phases (white) on the experimental phase diagram (Fig. 3a) with the liquid phase on the theoretical phase diagram (Fig. 3c).

On long timescales, there are the three possible outcomes: the C, H or A states. The phase separation between the H and A states clearly shows that there are no states with intermediate polaron density (see Supplementary Note 1 (Supplementary Fig. 3a, e), and Ref. [17]). The model, on the other hand, shows another crystalline state with 1/12 filling and the corresponding

domain states with fillings close to 1/12 that are not observed experimentally, even though their filling lies directly between the observed photoinduced H (1/12.4) and A (1/11) state. Possibly, the 1/12 superlattice is less stable and may only exist as a transient phase, but so far it has not been observed on any timescale. Although the photoexcitation fluence can be continuously changed, and the number of initially photoexcited carriers can be assumed to be directly proportional to the laser fluence, after the system self-organizes, only certain fillings are stable minima in the free energy, resulting in the observable phases. It is conceivable that the 1/12 phase does not have a sufficiently deep minimum in the free energy to make it stable enough to be observed as a properly identifiable phase. Possibly, its stability is hindered by the nearby presence of the 1/13 commensurate and H phases which are enhanced by Fermi surface nesting[42].

We note that in spite of its simplicity, the generic model reproduces the morphology of the observed metastable charge configurations. However, the model does not consider material-dependent features, such as Fermi surface nesting (which is relevant for 1$T$-TaS$_2$), the electron–phonon interaction that causes polaron formation, orbital overlap, or interlayer coupling, which was shown to play a role in the CDW stability[40]. We thus cannot expect it to predict with precision which of the many possible charge-ordered states could appear in a certain material.

To predict the temporal stability of the states at different photodoping values one needs to consider additional mechanisms which are beyond the scope of the model, such as long-range order and topological defects created in the H state transition for example[16]. Larger densities of such defects created by higher fluences may be the cause of the higher long-term stability of the metastable H phase at higher fluences, which was observed at 77 K. The amorphous state on the other hand is stabilized by the constraints imposed by the jamming, which is an entirely different and poorly understood stabilization mechanism[17].

We conclude that by a carefully chosen combination of techniques, phase-diagram snapshots can be obtained during the relaxation trajectory of a non-equilibrium system. Comparison of the experimental phase diagram with the theory under the assumption that photoexcitation is related to doping confirms that three out of four phases can be reached with photoexcitation fluence as the only control parameter. While the transition outcome reproducibility is excellent on the low-fluence side (up to ~3 mJ/cm$^2$), various factors that are not under direct control contribute to variable transition outcomes on the high fluence side. This has important implications for the stability and reliability of potential devices based on this material. The understanding of the time-evolution of different nonequilibrium states revealed by temporal phase diagrams opens the way to the development of new functionalities in metastable quantum materials based on configurational electron ordering.

## Methods

**Sample preparation**. 1T-TaS$_2$ crystals were grown by chemical transport method with iodine as a transport agent. The samples have average dimensions of $2 \times 2 \times 0.1$ mm. For optical measurements, the samples were glued to a copper plate with thermally conductive vacuum glue and cleaved just before inserting into the cryostat. For STM measurements, the samples were glued to the holder with a UHV compatible silver paste and inserted into the machine. The samples were cleaved inside the UHV chamber to prevent surface contamination.

**STM experiments**. We use a low-temperature STM (LT Nanoprobe, Scienta-Omicron) with optical access. For the photoexcitation of the samples, we use a 100 kHz, 800 nm laser system, with 50 fs pulses. The number of shots was selected using an acousto-optic modulator. The laser pulses are guided into the STM chamber using an automatic stabilizing system with a positioning precision of <5 µm. The laser beam profile was carefully determined externally with a CCD camera. A scanning electron microscope mounted above the STM allowed for precise tip positioning. The center position of the laser beam was determined from the perimeter defined by the border between the equilibrium and switched states.

**Femtosecond spectroscopy**. For the optical experiments, we use a 1 kHz, 800 nm laser system with 50 fs pulses. The sample is held in a vacuum cryostat with optical access. As an addition to the standard pump (P) and probe (p) pulses, we use the third driving (D) pulse to drive the sample out of equilibrium. We varied the fluence of the D pulse from 0.2 mJ cm$^{-2}$ to 10 mJ cm$^{-2}$. The fluences of the P and p pulses were always below 0.1 mJ cm$^{-2}$ to ensure minimal disturbance. We use a mechanical chopper to modulate the P pulse, while the D and p pulses are unmodulated. This makes the D pulse invisible in the repetitive measurement with the lock-in technique and we only observe its effect on the sample.

**Theoretical model**. The model is based on the charged lattice gas (CLG) Hamiltonian $H = \sum_{ij} V(i,j) n_i n_j$ where $n_i$ is the occupational number of a polaron at site $i$ with values either 0 or 1 and $V(i,j) = V_0 \exp(-r_{ij}/r_s)/r_{ij}$ is the Yukawa potential that describes the screening. $V_0 = e^2/\epsilon_0 a$ in CGS units and $r_i = |r_i|$ is the dimensionless position of the $i$th polaron and $r_s$ is the dimensionless screening radius. The dimensions are normalized so that the value 1 for both $|r_i|$ and $r_s$ corresponds to one lattice constant $a$. The value of $r_s$ is set to 4.5, which gives the best match with the experiments (for details of calculations with other values of $r_s$ see Ref. [25]). $e$ and $\epsilon_0$ are the electron charge and static dielectric constant of the material respectively. Polarons can only occupy the sites of the underlying triangular lattice. The ratio of polarons in the system divided by the total number of lattice sites is expressed as the filling $f$. The energy minimum is found by a Monte-Carlo method. We studied the phase diagram of such a system at fixed values of $f$ and $r_s$. The CLG interpretation of polarons assumes a system of interacting phonons and repulsive electrons that is canonically transformed into a system of interacting small polarons in the strong electron–phonon coupling limit. We neglect spin effects, assume a screened Coulomb interaction, and assume that the hopping of polarons $\tilde{t} \ll V_0$, which is justified by the static nature of observed charges and set to zero in this model.

## Data availability

All of the data supporting the conclusions are available within the article and the Supplementary Information. Additional data are available from the corresponding author upon reasonable request.

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

## Acknowledgements
The work was supported by ERC ADG Trajectory (GA320602) and the Slovenian Research Agency (project P10040 and young researcher grants, P17589 and P08333).

This project has received funding from the European Union's Horizon 2020 research and innovation program under the Marie Skłodowska-Curie grant agreement No. 701647. We would like to thank Petra Šutar and Aleš Mrzel for sample growth.

## Author contributions
J.R., I.V., Y.G., T.M., and D.M. conceived the experiments. J.R., M.D., Y.G., Y.V., and I.V. conducted the STM measurements and analyzed the data. J.R., I.V., and T.M. conducted the ultrafast optical measurements and analyzed the data. J.V. performed the theoretical calculations. J.R. and D.M. wrote the paper.

## Competing interests
The authors declare no competing interests.
