## [Peer Review File · Nature Communications]

REVIEWER COMMENTS

Reviewer #1 (Remarks to the Author):

* Referee report on manuscript NCOMMS-20-43928

"A time-domain phase diagram of metastable states in a charge ordered quantum material"

by Jan Ravnik, Michele Diego, Yaroslav Gerasimenko, Yevhenii Vaskivskyi, Igor Vaskivskyi, Tomaz Mertelj, Jaka Vodeb and Dragan Mihailovic

Submitted for consideration to Nature Communications.

** Summary and recommendation

Dear Editor,

In this manuscript the authors map out the temporal phase-boundaries of several electronic phases in the (quasi) two dimensional dichalcogenide 1T-TaS₂. Optical pump-probe reflectivity measurements are used to track the evolution of coherent phonon excitations in combination with a mapping of the morphology of the electronic surface density at low temperature using scanning tunneling microscopy. In particular the low temperature insulating commensurate charge density wave state and the metastable hidden metallic state is investigated.

The experimental results are complemented by theoretical modeling of the polaron morphology as a function of density, using a classical charged-lattice-gas model with immobile charged polarons on a triangular lattice, interacting with a long-range Yukawa potential.

I find the experimental design as beautiful as it is ingenious and the manuscript is very well written. I think that the subject matter of the manuscript is interesting to a wide audience and deserves the attention that a publication in Nature Communications would provide. However, there are a number of points where I think the manuscript could be improved and clarified.

Since I am not competent to comment on the experimental intricacies I have here focused on the presentation, interpretation, and connection to theory. In particular, I think the fact that only three distinct polaron fillings are observed experimentally deserves an in-depth discussion in the main text. Also the discussion of and connection to the theoretical modeling could be made more precise, please see the detailed comments below.

Sincerest regards,

Hugo U.R. Strand

** Comments

*** Missing discussion of discrete doping values in the main text

Regarding the experimentally observed fillings f , the authors state in the discussion of the theoretical simulations in the supplemental material (lines 138 - 140)

"In the experiment, the only values of f observed are $1/13$ in the commensurate state, $\sim 1/12.6$ in the hidden state and $\sim 1/11$ in the amorphous state."

In my opinion, this central experimental fact deserves to be clearly stated and discussed in the main text.

The authors attribute the absence of experimental observations of the continuous range of theoretically possible fillings to shortcomings of the simple theoretical model (lines 143 - 145)

"The absence of observation of all the possible fillings between the values $1/12.6$ and $1/11$, which includes the $1/12$ lattice as well as the mixture of $1/12$ and $1/13$ lattices is most likely due to various effects, which are not included in the model."

On this point I think the manuscript deserves to be extended with a discussion of what types of excitations the photo doping creates in the system, preferably also in the main text.

Why are there only a few special polaron fillings? Assuming that the pump fluence is proportional to the amount of excited electron-hole pairs, what are the other possible excitation channels, other than polaron formation?

*** Assumption of correspondence between incident photon density and electron/polaron filling

The main text is written under the assumption that the photon density and the induced polaron filling are directly related, e.g. on lines 236 - 238 the authors states

"assuming a correspondence between the photo-excited carrier density (which is proportional to incident photon density) and electron filling."

from line 243 "(1 polaron equals 1 electron)"

I find this at odds with the fact that only three specific polaron fillings/densities are observed experimentally, see my previous comment.

*** Presentation of the theoretical charged-lattice-gas model

While the charged-lattice-gas model is named in the introduction (line 67) the later explanation of the model (from line 231 and on) discuss it only in terms of "electrons".

I find the current formulations potentially a bit misleading and would suggest the following:

Change

Lines 233 - 234, "The model considers the ordering of electrons subject to screened Coulomb interaction on a triangular atomic lattice, ..."

to

"The model considers the ordering of **immobile polarons** subject to screened Coulomb interaction on a triangular lattice, ..."

Change

Lines 238 - 239, "The model defines the filling of the system as the number of electrons at the Fermi level divided by the number of atoms."

to

"The model defines the filling of the system as the number of polarons per unit cell."

(Please note that the model does not contain any (itinerant) electrons, and no Fermi level.)

***** Theoretically predicted liquid-phase not observed experimentally?**

At low temperature the charge-lattice-gas model predicts multiple phases in the relevant range of polaron fillings. The authors point out that the ordered phase at $1/12$ filling and the domain-wall phase emanating from this ordered phase is not observed experimentally.

The remaining three theoretical low temperature phases, however, are observed experimentally in the sense that the spatial morphology agrees with the STM data.

Here I find the manuscript lacking, since there seems to be no discussion of the theoretical high temperature liquid phase. I think a comment on this third, un-observed but theoretically predicted, phase is in place.

***** Predictive power of the theoretical model**

While the theoretical modeling using the charged-lattice-gas model is an important part of the manuscript, I think its predictive power is limited.

In the relevant polaron filling range, three out of the six theoretically predicted phases are observed experimentally.

Only three distinct polaron fillings are observed in the experiments as the incident photon density is varied continuously. Hence, it is not possible to compare the theoretical trends of the phase boundaries as a function of temperature and filling with the experimental phase boundaries in the plane of temperature and photon fluence.

For these reasons I recommend that the wording on the prowess of the model be tuned down.

Lines 251 - 252, "In spite of the crudeness of the simple model, which correctly predicts the observed metastable charge configurations, ..."

please consider changing "correctly predicts" to "reproduces the morphology of"

** Minor comments

*** Lines 40 - 43

Please give references on "the fundamentally different dynamical ordering phenomena" (apart from the Kibble-Zurek mechanism) e.g. the Ehrenfest scheme, topological, and jamming transitions.

*** Line 243

Should "unit area" be "unit cell" ?

*** Line 296

The electronic charge "e" is not defined in the text.

*** Line 303

Please specify the fixed interaction value r_s used in the calculations.

*** Line 307

Please point out that the hopping of the polarons is explicitly set to zero in the theoretical model.

*** Fig. 2b and 2d

Please increase the numerical frequency resolution in the Fourier spectra of Fig. 2b and 2d, using e.g. zero padding before Fourier transforming the time series.

*** Fig. 3c

Please show the temperature scale in the theoretical phase diagram.

Reviewer #2 (Remarks to the Author):

This paper reports on a time-resolved study of the charge-ordered material 1T-TaS₂. This compound is known to exhibit a rich phase diagram as a function of temperature and pressure, and to respond strongly to laser pulses excitations.

By a combination of STM and pump-probe measurements, the authors present an out-of-equilibrium phase diagram of 1T-TaS₂, as a function of time and photon density of the laser excitation.

This phase diagram is then compared to the theoretically calculated one, assuming the equivalence between doping and laser excitation. Except for one phase (so-called 1/12) there is a qualitative agreement between observed (Commensurate, Hidden, Amorphous phases) and calculated phases.

This is an interesting paper, which gives novel information on transient states observed in 1T-TaS₂, for the first time presented on a phase diagram, worth publishing in Nature Communication.

The results are sound and convincing and will help researchers in that field.

Below are some remarks and questions to improve the paper.

For the paper to be stand-alone, I recommend the authors to describe in more details the many phases of this compound (so-called C, NC, H, T, etc.). It is clear that this work is the last episode of a long (and interesting) story, and I encourage the authors not to consider the readers as experts of this very specific quantum material.

It is not clear how the "1H polytype" phase is characterized.

In the same vein, the interlayer structure is not taken into account in the discussion. Stahl 2020 et al. (ref. 37) have shown by X-ray measurements that the interlayer coupling is important for the stabilization of the H-phase. This should be discussed.

In fig. 3C, the different grey parts should be described.

Fig 2: please use F instead of Phi, consistently with the text..

Reviewer #3 (Remarks to the Author):

The manuscript by Ravnik et al. addresses the laser-induced manipulation of charge-density wave states in 1T-TaS₂, a material that has attracted great interest in recent years. One of the areas of very active research has been the so-called hidden state, a metastable metallic phase prepared by pulsed laser illumination of the low-temperature commensurate CDW phase. Discovered by the Mihailovic group, besides its fundamental appeal, this phase is considered a candidate for various applications involving reversible switching of the resistivity. However, the precise conditions for its formation and the relation to other nearby metastable phases are still uncertain.

Ravnik et al. attempt to resolve this issue in a systematic manner, combining scanning tunneling microscopy and ultrafast optical spectroscopy for a range of excitations and starting conditions, including base temperature, laser fluence and applied pulse number. The findings are cast into what the authors call 'time-domain phase diagrams', which illustrate the evolution of the respective phases as a function of time. The results are compared with recent model simulations.

The authors differentiate between the commensurate phase, the hidden state, the jammed state, and permanent structural modifications into the 1H state or more irregular structures. The key findings are summarized in the third figure, which displays observed phases at early times (fs-ps), intermediate times (ms) and quasi-persistent states (>1000s). The most reproducible finding is the hidden state, which is found both for single pulse illumination and pulse sequences. Among other results, it is shown that the hidden state is consistently reached at low temperature and low fluence, while the state is reached at later times for higher fluence.

I very much enjoyed reading the manuscript. It is a well-written piece and, together with the very valuable supplement, contains the most comprehensive presentation of the laser-prepared structural state its coherent amplitude-mode properties. Moreover, the work presents a novel perspective on the underlying phase categorization and necessary conditions, which will be instrumental for other research groups.

I am generally strongly in favor of publication in Nature Communications, but would suggest that the authors consider the following comments in a revision:

- 1) Perhaps the most important question concerns the relation to other works. I am somewhat familiar with the literature on the subject, although certainly not in every point of detail. I have not seen a comprehensive presentation of this type and find it exceedingly helpful. There is also clearly a large amount of original data shown. Yet, certain aspects, such as the temperature-dependent lifetime of the hidden state, and other points described by the authors, have been considered in detail before. I am therefore wondering if the authors can stress more clearly, which aspects of this work could not have been extracted and pieced together by prior findings. In other words, can the authors argue why this is not a comprehensive review with a re-interpretation of prior individual findings?

2) I personally have no problem in the loose notion of a time-domain phase diagram, but I assume others will object. It would therefore be useful if the authors discussed in some more detail the role of phase diagrams in equilibrium scenarios. Moreover, I believe it should be acknowledged that under non-equilibrium conditions, one cannot expect to present a universal diagram, but rather something that is heavily influenced by experimental conditions, such as the quality of thermal and mechanical contacts, film thicknesses and macroscopic sample geometry, etc.

3) There are some smaller typographical errors. Examples: Usage of Its' and It's when its was meant; the maximum fluence for experiments is noted as 12 mJ/cm² in most cases, but 22 mJ/cm² in the caption of Fig.1.

RESPONSE TO REVIEWER COMMENTS

(Author's remarks are in red)

Reviewer #1 (Remarks to the Author):

* Referee report on manuscript NCOMMS-20-43928

"A time-domain phase diagram of metastable states in a charge ordered quantum material"

by Jan Ravnik, Michele Diego, Yaroslav Gerasimenko, Yevhenii Vaskivskiy, Igor Vaskivskiy, Tomaz Mertelj, Jaka Vodeb and Dragan Mihailovic

Submitted for consideration to Nature Communications.

** Summary and recommendation

Dear Editor,

In this manuscript the authors maps out the temporal phase-boundaries of several electronic phases in the (quasi) two dimensional dichalcogenide 1T-TaS₂. Optical pump-probe reflectivity measurements are used to track the evolution of coherent phonon excitations in combination with a mapping of the morphology of the electronic surface density at low temperature using scanning tunneling microscopy. In particular the low temperature insulating commensurate charge density wave state and the metastable hidden metallic state is investigated.

The experimental results are complemented by theoretical modeling of the polaron morphology as a function of density, using a classical charged-lattice-gas model with immobile charged polarons on a triangular lattice, interacting with a long-range Yukawa potential.

I find the experimental design as beautiful as it is ingenious and the manuscript is very well written. I think that the subject matter of the manuscript is interesting to a wide audience and deserves the attention that a publication in Nature Communications would provide. However, there are a number of points where I think the manuscript could be improved and clarified.

Since I am not competent to comment on the experimental intricacies I have here focused on the presentation, interpretation, and connection to theory. In particular, I think the fact that only three distinct polaron fillings are observed experimentally deserves an in-depth discussion in the main text. Also the discussion of and connection to the theoretical modeling could be made more precise, please see the detailed comments below.

Sincerest regards,
Hugo U.R. Strand

AUTHORS: We wish to thank the reviewer for the kind remarks and particularly the detailed comments. We agree that the connection between the experiment and theory can be made more precise. We have

taken into account all of the suggestions and modified the paper according to the comments. We believe the paper has significantly improved in this way.

** Comments

*** Missing discussion of discrete doping values in the main text

Regarding the experimentally observed fillings ν , the authors state in the discussion of the theoretical simulations in the supplemental material (lines 138 - 140)

"In the experiment, the only values of ν observed are $1/13$ in the commensurate state, $\sim 1/12.6$ in the hidden state and $\sim 1/11$ in the amorphous state."

In my opinion, this central experimental fact deserves to be clearly stated and discussed in the main text.

AUTHORS: We agree that this is a statement that should appear also in the main text. We have extended the discussion to state this more explicitly, also in line with the further comments about the shortcomings of the model in relation to experiment (see later).

The authors attribute the absence of experimental observations of the continuous range of theoretically possible fillings to shortcomings of the simple theoretical model (lines 143 - 145)

"The absence of observation of all the possible fillings between the values $1/12.6$ and $1/11$, which includes the $1/12$ lattice as well as the mixture of $1/12$ and $1/13$ lattices is most likely due to various effects, which are not included in the model."

On this point I think the manuscript deserves to be extended with a discussion of what types of excitations the photo doping creates in the system, preferably also in the main text.

AUTHORS: The point was addressed by adding a discussion on p. 11.

Why are there only a few special polaron fillings? Assuming that the pump fluence is proportional to the amount of excited electron-hole pairs, what are the other possible excitation channels, other than polaron formation?

AUTHORS: This is a very good point and one of the big issues in all similar materials. There are other effects that one needs to take into account, such as orbital overlap, electron-phonon coupling and Fermi surface nesting (e.g. for $1/13$), which favour specific polaron fillings. The additional stabilization mechanisms are now discussed in more detail in the revised MS. Spin channels are thought not to be relevant in this material and are not discussed in the manuscript. Regarding other excitation channels, we have added a new paragraph with a discussion on the photoexcited electron-hole thermalization pathways and other excitation channels on p. 11.

*** Assumption of correspondence between incident photon density and electron/polaron filling

The main text is written under the assumption that the photon density and the induced polaron filling are directly related, e.g. on lines 236 - 238 the authors states

"assuming a correspondence between the photo-excited carrier density (which is proportional to

incident photon density) and electron filling."

from line 243 "(1 polaron equals 1 electron)"

I find this at odds with the fact that only three specific polaron fillings/densities are observed experimentally, see my previous comment.

AUTHORS: Again, a very good point, related to the previous comment. The photoexcitation fluence can be continuously changed. The number of initially photoexcited carriers is assumed to be directly proportional to the laser fluence and is also continuous. However, after the system self-organizes, only certain fillings are stable minima in the free energy, resulting in the observable phases. It is conceivable that the 1/12 phase also exists on some intermediate timescale, but we do not observe it, suggesting that it does not have a sufficiently deep minimum in the free energy to make it a stable long-range ordered phase. The possible reasons for this are discussed in the revised MS (e.g. e-p coupling, nesting, inter-layer coupling etc.)

*** Presentation of the theoretical charged-lattice-gas model

While the charged-lattice-gas model is named in the introduction (line 67) the later explanation of the model (from line 231 and on) discuss it only in terms of "electrons".

I find the current formulations potentially a bit misleading and would suggest the following:

Change

Lines 233 - 234, "The model considers the ordering of electrons subject to screened Coulomb interaction on a triangular atomic lattice, ..."

to

"The model considers the ordering of **immobile polarons** subject to screened Coulomb interaction on a triangular lattice, ..."

Change

Lines 238 - 239, "The model defines the filling of the system as the number of electrons at the Fermi level divided by the number of atoms."

to

"The model defines the filling of the system as the number of polarons per unit cell."

(Please note that the model does not contain any (itinerant) electrons, and no Fermi level.)

AUTHORS: We agree with the comments. We implemented the changes as suggested.

*** Theoretically predicted liquid-phase not observed experimentally?

At low temperature the charge-lattice-gas model predicts multiple phases in the relevant range of polaron fillings. The authors point out that the ordered phase at $1/12$ filling and the domain-wall phase emanating from this ordered phase is not observed experimentally.

The remaining three theoretical low temperature phases, however, are observed experimentally in the sense that the spatial morphology agrees with the STM data.

Here I find the manuscript lacking, since there seems to be no discussion of the theoretical high temperature liquid phase. I think a comment on this third, un-observed but theoretically predicted, phase is in place.

AUTHORS: We would like to thank the reviewer for pointing this out. We really have missed the opportunity to discuss the high temperature phase. We have added a paragraph describing this phase and linking it to the unidentified out of equilibrium phases on the short timescales. In particular, we discuss the process of formation of polarons into a liquid, and their subsequent self-organization with reference to experimental ARPES data and the coherent response of the polaronic amplitude mode that shows the sequence of events in polaron formation.

*** Predictive power of the theoretical model

While the theoretical modeling using the charged-lattice-gas model is an important part of the manuscript, I think its predictive power is limited.

In the relevant polaron filling range, three out of the six theoretically predicted phases are observed experimentally.

Only three distinct polaron fillings are observed in the experiments as the incident photon density is varied continuously. Hence, it is not possible to compare the theoretical trends of the phase boundaries as a function of temperature and filling with the experimental phase boundaries in the plane of temperature and photon fluence.

For these reasons I recommend that the wording on the prowess of the model be tuned down.

AUTHORS: We agree that the limitations of the model should be more clearly stated.

Lines 251 - 252, "In spite of the crudeness of the simple model, which correctly predicts the observed metastable charge configurations, ..."

please consider changing "correctly predicts" to "reproduces the morphology of"

AUTHORS: We agree with the remark and have changed the text accordingly, including the suggested text. We note that in response to previous issues, we have also added the discussion on the possible reasons why the model fails, at $f=1/12$ and the 'hidden' and amorphous phases nearby that accompany it.

** Minor comments

*** Lines 40 - 43

Please give references on "the fundamentally different dynamical ordering phenomena" (apart from the Kibble-Zurek mechanism) e.g. the Ehrenfest scheme, topological, and jamming transitions.

AUTHORS: We agree and have added the appropriate citations where needed.

*** Line 243

Should "unit area" be "unit cell" ?

AUTHORS: We agree that unit cell is a better term, and have changed it.

*** Line 296

The electronic charge "e" is not defined in the text.

AUTHORS: We would like to thank the reviewer for pointing this out. We have added the definition.

*** Line 303

Please specify the fixed interaction value r_s used in the calculations.

AUTHORS: We have specified the screening radius r_s in the methods section.

*** Line 307

Please point out that the hopping of the polarons is explicitly set to zero in the theoretical model.

AUTHORS: We added the point to the description.

*** Fig. 2b and 2d

Please increase the numerical frequency resolution in the Fourier spectra of Fig. 2b and 2d, using e.g. zero padding before Fourier transforming the time series.

AUTHORS: We have increased the resolution of Fourier transforms as suggested.

*** Fig. 3c

Please show the temperature scale in the theoretical phase diagram.

AUTHORS: We have added the temperature scale to the theoretical phase diagram, as suggested.

Reviewer #2 (Remarks to the Author):

This paper reports on a time-resolved study of the charge-ordered material 1T-TaS₂. This compound is known to exhibit a rich phase diagram as a function of temperature and pressure, and to respond strongly to laser pulses excitations.

By a combination of STM and pump-probe measurements, the authors present an out-of-equilibrium phase diagram of 1T-TaS₂, as a fonction of time and photon density of the laser excitation.

This phase diagram is then compared to the theoretically calculated one, assuming the equivalence between doping and laser excitation. Except for one phase (so-called 1/12) there is a qualitative agreement between observed (Commensurate, Hidden, Amorphous phases) and calculated phases.

This is an interesting paper, which gives novel information on transient states observed in 1T-TaS₂, for the first time presented on a phase diagram, worth publishing in Nature Communication.

The results are sound and convincing and will help reseachers in that field.

AUTHORS: We would like to thank the reviewer for the kind remarks. We have looked through the comments and made the desired changes.

Below are some remarks and questions to improve the paper.

For the paper to be stand-alone, I recommend the authors to describe in more details the many phases of this compound (so-called C, NC, H, T, etc.). It is clear that this work is the last episode of a long (and interesting) story, and I encourage the authors not to consider the readers as experts of this very specific quantum material.

AUTHORS: We would like to thank the reviewer for the sensible comment. In the revised MS, we have added more details about the structure of 1T-TaS₂ and its different equilibrium states. We have also added a more detailed description of the 1H polytype.

It is not clear how the "1H polytype" phase is characterized.

In the same vein, the interlayer structure in not taken into account in the discussion. Stahl 2020 et al. (ref. 37) have shown by X-ray measurements that the interlayer coupling is important for the stabilization of the H-phase. This should be discussed.

AUTHORS: Regarding the 1H polytype, we have added a description in the introduction, as was suggested in the previous remark. We have also added a sentence to the experimental part, saying "*The 1H polytype can be clearly distinguished from the 1T polytype based on the lack of charge ordering above 70 K and by a 3x3 CDW at lower temperature*". We also agree that interlayer structure is important to consider in this material. We have added the discussion about the interlayer coupling to the part where we discuss the model and the reasons for the stability of the H and other phases.

In fig. 3C, the different grey parts should be described.

AUTHORS: We would like to thank the reviewer for this remark. The grey parts of the theoretical phase diagram in Fig. 3C refer to the theoretical 'liquid'. In response we have added a paragraph discussing this part of the theoretical phase diagram, connecting it to the (white) regions on the experimental phase diagram in Fig 2a and b. its morphology is also shown in the legend (bottom figure), whereby the density varies according to the filling. (In the revised MS the grey was changed to green color for clarity, It was previously a bit misleading, because it was the same color as the ISC part in the experimental phase diagram).

Fig 2: please use F instead of Phi, consistently with the text..

AUTHORS: We thank the reviewer for pointing this out. We made the change to the Figure.

Reviewer #3 (Remarks to the Author):

The manuscript by Ravnik et al. addresses the laser-induced manipulation of charge-density wave states in 1T-TaS₂, a material has attracted great interest in recent years. One of the areas of very active research has been the so-called hidden state, a metastable metallic phase prepared by pulsed laser illumination of the low-temperature commensurate CDW phase. Discovered by the Mihailovic group, besides its fundamental appeal, this phase is considered a candidate for various applications involving reversible switching of the resistivity. However, the precise conditions for its formation and the relation to other nearby metastable phases are still uncertain.

Ravnik et al. attempt to resolve this issue in a systematic manner, combining scanning tunneling microscopy and ultrafast optical spectroscopy for a range of excitations and starting conditions, including base temperature, laser fluence and applied pulse number. The findings are cast into what the authors call 'time-domain phase diagrams', which illustrate the evolution of the respective phases as a function of time. The results are compared with recent model simulations.

The authors differentiate between the commensurate phase, the hidden state, the jammed state, and permanent structural modifications into the 1H state or more irregular structures. The key findings are summarized in the third figure, which displays observed phases at early times (fs-ps), intermediate times (ms) and quasi-persistent states (>1000s). The most reproducible finding is the hidden state, which is found both for single pulse illumination and pulse sequences. Among other results, it is shown that the hidden state is consistently reached at low temperature and low fluence, while the state is reached at later times for higher fluence.

I very much enjoyed reading the manuscript. It is a well-written piece and, together with the very valuable supplement, contains the most comprehensive presentation of the laser-prepared structural state its coherent amplitude-mode properties. Moreover, the work presents a novel perspective on the underlying phase categorization and necessary conditions, which will be instrumental for other research groups.

I am generally strongly in favor of publication in Nature Communications, but would suggest that the authors consider the following comments in a revision:

AUTHORS: We are grateful for the review and the insightful remarks. We made the changes to the manuscript accordingly.

1) Perhaps the most important question concerns the relation to other works. I am somewhat familiar with the literature on the subject, although certainly not in every point of detail. I have not seen a comprehensive presentation of this type and find it exceedingly helpful. There is also clearly a large amount of original data shown. Yet, certain aspects, such as the temperature-dependent lifetime of the hidden state, and other points described by the authors, have been considered in detail before. I am therefore wondering if the authors can stress more clearly, which aspects of this work could not have been extracted and pieced together by prior findings. In other words, can the

authors argue why this is not a comprehensive review with a re-interpretation of prior individual findings?

AUTHORS: We think this is a very good point and we will be glad to clear it. Previously, the discovery of different phases was reported by different, unconnected methods under very different experimental conditions. The regions of stability were not systematically investigated. Moreover, the experimental conditions could hardly be compared. The temporal time evolution was not considered previously at all in the context of a temporally-evolving phase diagram that spans more than 12 orders of magnitude. This is conceptually new. The approach presented here reveals a new insight from a viewpoint not previously considered.

The present work systematically delineates the phase boundaries addressing the question whether there is a sharp boundary between phases, or the domain wall density smoothly increases with increasing fluence. There is nothing in present theories that can be used as a guideline if such a transition exists, or where the boundary is.

Since there are no systems that have been reported to display such behavior, it is paramount to charter the phase diagram in as much detail as necessary to ascertain the boundaries of the different phases. We also note that the method of using the entire area exposed by a single shot for analysis of different fluences is novel.

There are also substantial new data, systematically changing the fluence across the whole region of the phase diagram at different temperatures (previously only a few rather arbitrary points were taken from which the boundaries between phases could not be ascertained). In addition: (i) 4K and 77K STM data over a large range of fluence are new, (ii) we show for the first time that the outcome for the H state is the same irrespective of the number of pulses, (iii) the region of high fluence was never investigated, particularly near damage threshold.

2) I personally have no problem in the loose notion of a time-domain phase diagram, but I assume others will object. It would therefore be useful if the authors discussed in some more detail the role of phase diagrams in equilibrium scenarios. Moreover, I believe it should be acknowledged that under non-equilibrium conditions, one cannot expect to present a universal diagram, but rather something that is heavily influenced by experimental conditions, such as the quality of thermal and mechanical contacts, film thicknesses and macroscopic sample geometry, etc.

AUTHORS: We would like to thank the reviewer for this important remark.

What we usually call equilibrium is in fact described on an anthropocentric time scale, which means the system does not change much on a timescale of days, weeks or a few years, i.e. over a timescale of 10^4 to 10^8 seconds. This range of timescales is actually small compared to the 12 orders of magnitude that we are trying to present here. Our premise is that we need to make sure that the measurement time-window is short enough to perceive it as static (thus quasi equilibrium) on that respective time scale. This is in fact realized in all three cases. On the picosecond timescale, the measurements take tens of picoseconds, but we have shown that the behavior does not change for >400 ps, on the millisecond timescale, we measure with the same method, which is 8 orders of magnitude faster than the time after the excitation. STM experiments are so slow, that we measure only the long-term "equilibrium" states. We have added some additional discussion about phase diagrams to the introduction. A short discussion was added about the effect of experimental conditions, sample mounting, geometry etc., that influences the time-axis, as suggested.

3) There are some smaller typographical errors. Examples: Usage of Its' and It's when its was meant; the maximum fluence for experiments is noted as 12 mJ/cm^2 in most cases, but 22 mJ/cm^2 in the caption of Fig.1.

AUTHORS: We would like to thank the reviewer for pointing those mistakes out. Regarding Fig. 1b, the fluence actually was higher than in the rest of the experiments, as we used a different and much simpler experimental setup. We added a note in the MS to avoid confusion: "*SEM image of an area continuously exposed to 22 mJ/cm^2 peak fluence, showing damage at the center of the laser spot, which was used in the initial optical setup to define the beam position.*".

REVIEWERS' COMMENTS

Reviewer #1 (Remarks to the Author):

Dear Editor,

Thank you for sending me the revised version of manuscript NCOMMS-20-43928 under consideration for Nature Communications. The authors have addressed all the concerns I raised in my first report and have also considered the issues raised by the other referees.

Therefore, I am recommending the manuscript for publication in Nature Communications.

Best regards,

Hugo U.R. Strand

Reviewer #2 (Remarks to the Author):

The authors have satisfactorily answered my questions. Informative precisions have been added to the manuscript, worth publishing in Nature Communication.

Reviewer #3 (Remarks to the Author):

I have considered the response by the authors to my review and the others. I believe the authors have appropriately addressed all comments and questions, and I suggest publication of the manuscript.